# Haunted Oppressors: The Deconstruction of Manliness in the Imperial Gothic Stories of Rudyard Kipling and Arthur Conan Doyle

**Anna Berger**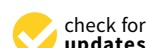

English Department, University of Tübingen, 72074 Tübingen, Germany; anna.berger@uni-tuebingen.de

**Abstract:** Building on Patrick Brantlinger's description of imperial Gothic fiction as "that blend of adventure story with Gothic elements", this article compares the narrative formula of adventure fiction to two tales of haunting produced in a colonial context: Rudyard Kipling's "The Mark of the Beast" (1890) and Arthur Conan Doyle's "The Brown Hand" (1899). My central argument is that these stories form an antithesis to adventure fiction: while adventure stories reaffirm the belief in the imperial mission and the racial superiority of the British through the display of hypermasculine heroes, Kipling's and Conan Doyle's Gothic tales establish connections between imperial decline and masculine failure. In doing so, they destabilise the binary construction between civilised Western self and savage Eastern Other and thus anticipate one of the major concerns of postcolonial criticism. This article proposes, therefore, that it is useful to examine "The Mark of the Beast" and "The Brown Hand" through a postcolonial lens.

**Keywords:** imperial Gothic; haunting; adventure fiction; hegemonic masculinity; postcolonial studies

## 1. Introduction

> [W]hen I tell you that for four years I have never passed one single night, either in Bombay, aboard ship, or here in England, without my sleep being broken by this fellow, you will understand why it is that I am a wreck of my former self. (Conan Doyle 2000, p. 78)

The above passage from Arthur Conan Doyle's "The Brown Hand", which was first published in *The Strand Magazine* in 1899, illustrates what can happen to men in the far-flung corners of the British empire: they might be reduced to a wreck of their former selves. In the story, Sir Dominick Holden, who acquired a reputation as "the most distinguished Indian surgeon of his day" (Conan Doyle 2000, p. 71), has to pay the price for his ignorance towards the people he treated. After his return to England, he is haunted by the ghost of an Afghan whose hand he had amputated during his time in India and kept as a fee for his services. The story implies that the agents of empire do not go unpunished. Foreign horrors might follow them back from the colonies.

Similarly, in Rudyard Kipling's "The Mark of the Beast", which was first published in the *Pioneer* in 1890, an Englishman is punished supernaturally for his disregard for the culture and religion of Hindu people in North India. The story features three Englishmen who live in Dharamsala.[1] One of them is cursed after having damaged a Hindu temple. The other two Englishmen fight for the soul of their comrade who degenerates into a liminal being between human and animal with a penchant for raw meat.

---

[1] Kipling spells the place "Dharmsala" (Kipling 2006, p. 3) in his story.

Both stories can be attributed to imperial Gothic fiction, a sub-genre of the Gothic which Patrick Brantlinger has described as "that blend of adventure story with Gothic elements" (Brantlinger 1988, p. 227). This article illustrates that it is, in fact, instructive to compare imperial Gothic fiction to Victorian and Edwardian adventure tales. Both genres situate questions of masculinity within an imperial context, but with a radically different outcome: while the colonial setting is used as a testing ground for the young and successful hero in adventure tales, it becomes the site of personal failure in imperial Gothic fiction.

What both genres have in common is that they present the foreign countries and their inhabitants as something dangerous that must be controlled. However, while the heroes of adventure fiction successfully exhibit what they perceive as their "God-given right, as well as the duty, to govern and control" (Kanitkar 1994, p. 194), the protagonists of imperial Gothic fiction often end up being controlled by the foreign culture. This is also the case in the two stories discussed here. Both narratives express the concern that the racially superior white man might be "contaminated by the alien world" (Byron and Punter 2004, p. 40) and regress to a barbaric state—a theme which Melissa Edmundson has identified as characteristic of stories by male writers of colonial Gothic fiction (Edmundson 2018, p. 8). The fear of "going native" (Brantlinger 1988, p. 230) reflects a quintessentially Western point of view in which the colonised subjects are characterised by their Otherness. Edward Said has pointed out that "European culture gained in strength and identity by setting itself off against the Orient" (Said 2003, p. 3). "The Mark of the Beast" and "The Brown Hand" replicate the construction of the Orient as exotic and potentially monstrous, thereby justifying Britain's imperialist politics.

However, it would be short-sighted to view "The Mark of the Beast" and "The Brown Hand" solely as a warning about the corrupting and potentially dehumanising influences of the foreign Other. The white colonisers are shown as responsible for their misery. Both stories are thus marked by an ambivalent treatment of the colonial encounter. In addition, it is the body of the white middle-class man that is othered and abnormalized: while Fleete regresses to a liminal being between human and animal, Holden is reduced to a ghost-like version of himself. In this way, both stories blur the binary divisions between civilised Western self and savage Eastern Other—dualisms which are clear-cut in imperial adventure fiction.

By destabilising the binary between Occident and Orient, civilised and savage, oppressor and oppressed, "The Mark of the Beast" and "The Brown Hand" anticipate one of the major concerns of postcolonial criticism. This article shows that the Gothic mode used in both stories facilitates these deconstructions. From its inception in the 18 century onwards, Gothic writing was used to transgress boundaries and break with societal rules. As Rosemary Jackson has observed, supernatural literature "explores and thereby threatens to dissolve many of the structures upon which social definitions of reality depend" (Jackson 1989, p. xviii). This tendency of the Gothic to expose "dualisms as well as the socially constructed hierarchies they mask" (Frye 1998, p. 169) lends itself well to postcolonial criticism. William Hughes and Andrew Smith are therefore right in their assertion that

> the Gothic is, and has always been, *post*-colonial, and this is where, in the Gothic text, disruption accelerates into change, where the colonial encounter—or the encounter which may be read or interpreted through the colonial filter—proves a catalyst to corrupt, to confuse or to redefine the boundaries of power, knowledge and ownership. (Hughes and Smith 2003a, p. 1)

Hughes and Smith identify two modes of postcolonial writing: writing that was produced "out of a postcolonial context" and "Gothic writing which was produced within a colonialist context, and which may be usefully interpreted in light of postcolonial ideas" (Hughes and Smith 2003b, p. 4). Similarly, James Procter and Angela Smith argue that texts written within the colonial period can "offer a critique of the empire from within" (Procter and Smith 2007, p. 97). In line with these arguments, this article proposes that it is illuminating to look at "The Mark of the Beast" and "The Brown Hand" through a postcolonial lens. Both stories use the supernatural in order to make visible,

criticise, and eventually reverse power structures.[2] Since both texts employ a Eurocentric worldview, it would be highly problematic, however, to attribute them completely to the postcolonial Gothic. I suggest, therefore, to situate them at the borderline between imperial Gothic and postcolonial Gothic. Both stories erode the dualism between white male coloniser and savage Other in that they invert the success-formula of the popular adventure tales and, in the process, make the shortcomings of white middle-class masculinity visible. In order to illustrate this, I will briefly pinpoint the narrative formula of imperial adventure fiction and the ideal of masculinity these stories promote in the following. In a second step, I will turn to Kipling's and Conan Doyle's stories and interrogate the ways in which these tales use the Gothic to counter adventure fiction and, in the process, reveal concerns about the rightfulness of Britain's imperial project.

## 2. The Construction of a Myth: Displays of Manliness in Imperial Adventure Fiction

The final decades of the nineteenth century witnessed the emergence of a new ideal of masculinity, which is commonly referred to as "imperial masculinity" (Beynon 2002; Deane 2014). Where earlier notions of manliness stressed personal development and moral maturation as exemplary virtues, this concept emphasised the importance of physical strength and mental toughness (Tosh 2005, p. 22). A striking aspect of this form of manliness was its rigorous rejection of emotionalism as a supposedly feminine character trait. The ideal was influenced by the British Empire insofar as it stressed qualities needed for combat and service like strength, psychological toughness and a will to win. Team sports, which formed an integral part of the school curriculum at that time, played a significant role in this context. It was believed that it would teach boys to subordinate themselves to the team effort and to obey or give orders (Mangan 2012, p. 60). Military heroes like General Robert Baden-Powell, who founded the "Boy Scout Movement" in 1907, furthered this connection by asserting that the playing field prepared boys and young men for the battlefield (Mangan 1998, pp. 47–48). The reality was of course more multifaceted than the ideal. Nevertheless, "there existed a discourse around this ideal" (Heholt and Parsons 2018, p. 15) which forged the idea of a hegemonic masculinity every man should conform to.

As one of the most widely consumed genres in the late nineteenth and early twentieth centuries (Boyd 2003, pp. 5–6, 34), imperial adventure tales contributed to the discourse about ideal masculinity. There is a particular narrative formula which writers of adventure fiction successfully exploited: The hero is usually a plucky and resourceful adolescent boy or young man who engages in a series of trials somewhere in the colonies in which he is able to establish and/or prove his manly qualities.

Many authors of adventure tales were not only highly self-conscious in their depiction of manliness, their stories were also "cheerleading for imperialism" (Carter 2014). Examples are almost too easy to find: In Robert Michael Ballantyne's successful *Coral Island* (1856), which relates the story of three boys who are shipwrecked in the South Pacific and subsequently marooned on an island, one of the heroes suggests to his comrades to take possession of the island by entering the service of its native inhabitants: "Of course we'll rise, naturally, to the top of affairs. White men always do in savage countries" (Ballantyne 1901, p. 27). The character of the white male adventurer is pitted against the supposedly inferior character of the native people. Ballantyne's contemporary George Alfred Henty was even more offensive in his tendency to exploit and reinforce racial stereotypes. In his writing, colonised people are repeatedly portrayed as "lazy, childlike, without capacity" (Arnold 1980, p. 79). In *By Sheer Pluck: A Tale of the Ashanti War* (1884), a narrative about General Garnet Wolseley's expedition against the Ashanti in West Africa, one of the characters describes the indigenous people as possessing an intelligence "equal to that of a European child of ten years old. ... Living among white

---

2   It is worth noting that supernatural literature is a sub-genre of Gothic fiction. The Gothic also encompasses narratives of the explained supernatural in which a rational explanation is provided for the seemingly supernatural events. With the exception of *Gaston de Blondeville* (1826), the novels of Anne Radcliffe belong to this sub-genre of the Gothic, for instance. In the stories discussed here, by contrast, the supernatural occurrences do not turn out to have natural causes.

men, their imitative faculties enable them to attain a considerable amount of civilization. Left alone to their own devices they retrograde into a state little above their native savagery" (Henty 1884, p. 188). By pointing out that they will remain in a state of savagery when left alone, the novel does not only render the native people inferior to white men but also justifies the extension of British rule and culture.

The two examples presented here are only two out of many. As Martin Green has remarked, adventure tales were, collectively, "the story England told itself as it went to sleep at night" (Green 1979, p. 3). As such, juvenile literature contributed to "the mass of formal and informal propaganda" (Richards 1989, p. 77) for Britain's imperial aspirations. By "celebrating what it meant to be English in the century when Britain dominated the world" (Boyd 2003, p. 45), imperial adventure tales served as a means to reinforce the belief in the racial and cultural superiority of the white middle-class male.

Just as in adventure fiction, the foreign environment in the colonies is often used as a space for the negotiation of questions of masculinity and nationhood in imperial Gothic writing. It is important to note, however, that imperial Gothic fiction does not mimic the narrative formula of adventure stories but works as a reversal of the genre.

### 3. Reverting to the Barbaric: Degenerate Englishmen in Colonial Gothic Fiction

*3.1. Rudyard Kipling: "The Mark of the Beast" (1890)*

The first imperial Gothic story discussed here is by Rudyard Kipling. It is difficult to imagine a writer more entangled with the British Empire than the Indian-born author and journalist. Among his Anglo-Indian contemporaries, he had "a reputation for knowing more about the colonial underworld than the police" (Arata 1996, p. 163). For British readers, his writings gave an insight into the distant culture and frame of mind of Indian people. As his contemporary Andrew Lang has put it with regards to one of Kipling's short story collections: "It may safely be said that *Plain Tales from the Hills* will teach more of India, of our task there, of the various peoples whom we try to rule, than many Blue Books" (Lang 1971, p. 48). By explaining the foreign Other to British readers, Kipling's writings fulfilled a significant cultural function that served the colonial mission. Naturally, this perception of Kipling as spokesman for the British imperial task has strongly influenced the perception of his writing. However, there is more to Kipling than "[t]his familiar image . . . as popular apologist for the dominant ideology" (Arata 1996, p. 151). Especially his early Gothic fiction betrays a more ambivalent attitude towards the imperial project. This ambivalence is evident in "The Mark of the Beast" (1890) which was initially rejected for publication in England in 1886 with the strong recommendation to "instantly . . . burn this detestable piece of work" (qdt. in Hamilton 1983, p. 133). This strong reaction is indicative of the story's potential to alienate and offend English reading audiences.

"The Mark of the Beast" is told by an unnamed homodiegetic narrator who is spending New Year's Eve with his friends Strickland, a policeman, and Fleete, who recently "owned a little money and some land in the Himalayas" (Kipling 2006, p. 3). At the beginning of the story, British India is presented as a place where Englishmen can celebrate their Englishness in an exclusively male environment: "On New Year's Eve there was a big dinner at the club, and the night was excusably wet. When men foregather from the uttermost ends of the Empire, they have a right to be riotous" (ibid.), the narrator relates. These introductory remarks imply that Strickland, Fleete and the narrator belong to a privileged group of middle-class men. They are "'clubbable' men, the type of empire builder who would no doubt establish a club at any convenient meeting point in the colonial world" (Hughes 2014, p. 12). Furthermore, the narrator excuses Fleete's behaviour later that evening by pointing out that the night is "excusably wet" and that they "have a right to be riotous".

On their way back from the club, Fleete runs up the stairs of a Hindu temple and offends native priests by "gravely grinding the ashes of his cigar-butt into the forehead of the red stone image of Hanuman[,] . . . the Monkey God" (Kipling 2006, p. 4). The drunken Englishman justifies the vandalism by alluding to a passage in Revelation, Chapter 13, in which the "Beast out of the Earth", which is

commonly identified as Anti-Christ, requires those who worship him to wear a mark on the right hand or the forehead: "Shee that? Mark of the B-beasht! *I* made it. Ishn't it fine?" (ibid.) By putting a mark on the statue's forehead, Fleete identifies Hanuman as a disciple of Anti-Christ and, in extension, implies that the local population worships a false prophet. In this way, he indirectly justifies the missionary idea of British imperialism. Fleete's branding further evokes connotations of branding as a mark of belonging. By branding the statue of the Hindu deity, Fleete draws attention to the status of India as British dominion and thereby reinforces his position as white coloniser. Fleete's actions are thus a demonstration of British superiority in more than one way: they highlight British supremacy and present Christianity as the only true religion.

Up to this point, "The Mark of the Beast" resembles an imperial adventure tale in which the particularly English character of the white male adventurer is pitted against the supposedly uncivilised character of the natives whose country he explores. However, Western superiority starts to be threatened when the supernatural invades the story. Fleete is cursed by a naked, faceless leper who emerges from behind the statue of Hanuman and, in revenge for his disrespect of the god, leaves a mark on Fleete's breast. In the following, the Englishman's character regresses to that of a beast:

> Fleete could not speak, he could only snarl, and his snarls were those of a wolf, not of a man. . . . We were dealing with a beast that had once been Fleete . . .
>
> We bound this beast with leather thongs of the punkah-rope, and tied its thumbs and big toes together, and gagged it with a shoehorn, which makes a very efficient gag if you know how to arrange it. (Kipling 2006, p. 9)

Fleete's transgression to a beast is reminiscent of a werewolf transformation. As Ardel Haefele-Thomas has remarked, "Fleete is no longer just a human or just an animal, but a hybridized combination of both" (Haefele-Thomas 2018, p. 103). Kathleen Spencer has observed that in the second half of the nineteenth century, "Darwinian evolutionary theory blurred the boundaries between human and animal in not one but two ways: by the famous argument that humans and apes had a common ancestor, but also by the implied hierarchy at the end of *The Descent of Man* which leads from the ape-like ancestor through primitive peoples to civilized Europeans" (Spencer 2004, p. 310). Within this hierarchy, native people occupy a liminal position between human and animal—a position which justified their often inhumane treatment under British rule. As a werewolf, Fleete similarly becomes a hybrid figure and thus loses his status as a human being. This development is underscored by a shift of the personal pronoun from "he" to "it" in the above passage.

By identifying the supernatural powers of the colonised Other as the source of Fleete's regression, the story highlights the supposed monstrosity of the indigenous culture. At the same time, Fleete's transformation indicates his susceptibility to racial decline. In this way, "The Mark of the Beast" mirrors fin-de-siècle anxieties about the degeneration of the British race. Darwin's evolutionary theory formed the basis for such concerns, "for if humans could evolve, it was thought that they could also *de*volve or degenerate, both as nations and individuals" (Spencer 2004, p. 311). Medical issues like the spread of venereal diseases, the monstrosity of the Whitechapel murders, and the scandals which revolved around a number of homosexuals in the latter decades of the nineteenth century were seen as proof for Britain's degeneration. The feeling was that, like the Roman Empire, the British Empire was rotting from within (Beynon 2002, p. 38). Fleete's brutish behaviour and his subsequent regress to a beast dwell on these fears. In this context, it is noteworthy that Fleete is bewitched by a man who worships the Hindu deity Hanuman, which is commonly depicted as a man with the face and tail of a monkey. By emphasising the supernatural power of the god's disciple, the story reverses the hierarchical order between human and ape.

In an attempt to cure Fleete, Strickland and the narrator consult a doctor, who falsely diagnoses the Englishman with rabies. Dr Dumoise's inability to help Fleete emphasises "the story's insistence upon the impotence of Western rationality as paradigm of civilisation and progress, and the power of

the native supernatural culture" (Generani 2016, p. 27). Eventually, Strickland and the narrator are forced to believe in the superiority of the foreign culture:

> Strickland told me, in a whisper, his suspicions. They were so wildly improbable that he dared not say them aloud; and I, who entertained all Strickland's beliefs, was so ashamed of owning to them that I pretended to disbelieve. (Kipling 2006, p. 10)

The Englishmen suspect that the leper has bewitched Fleete. However, they are ashamed of their suspicions, which suggests that they are aware that their acceptance of the foreign religious power reflects badly on their status as Western men. By demonstrating the superior power of the colonial Other, the narrative reveals British supremacy as illusory.

"The Mark of the Beast" repeatedly breaks with the distinction between savage Other and civilised self. This is particularly evident in the following passage, in which Strickland and the narrator try to force the leper into lifting the curse:

> Strickland wrapped a towel round his hand and took the gun-barrels out of the fire. I put the half of the broken walking stick through the loop of fishing-line and buckled the leper comfortably to Strickland's bedstead. I understood then how men and women and little children can endure to see a witch burnt alive; for the beast was moaning on the floor, and though the Silver Man had no face, you could see horrible feelings passing through the slab that took its place . . .
>
> Strickland shaded his eyes with his hands for a moment and we got to work. This part is not printed here. (Kipling 2006, p. 12)

The narrator's refusal to reveal the extent of their torture suggests that he is not willing to disclose the brutality of British rule in India. As Gustavo Generani has observed, "[t]here is no imperial civilising mission in those behaviours, but plain barbarism" (25). The narrator justifies their brutality by comparing the torture to the burning of a witch. The torture is thus presented as something they only "endure" because they are convinced of the rightfulness of their actions. The comparison further highlights the leper's similarities to a witch. Like a witch, he put a curse on a human being and is thus justly punished. By implying the monstrosity of the indigenous culture, the story emphasises the necessity of Britain's imperial mission and even presents torture as a legitimate means to control the demonic power of the Other. As Gustavo Generani has pointed out, "[i]t is a position founded on an implicit argument: that British barbarism is justified by the influence of a barbarian context" (Generani 2016, p. 30). Nevertheless, the passage betrays the men's brutality, for the narrator sees the Silver Man[3] express "horrible feelings". The narrator's assurance that he "buckled the leper comfortably to Strickland's bedstead" can therefore only be understood as crude sarcasm. It is also worth noting that Strickland and the narrator duplicate Fleete's crime by branding the leper with gun-barrels. This act of branding can be understood as an attempt to re-establish British superiority over the colonised Other.

Eventually Strickland and the narrator succeed in their mission and rid Fleete of the curse. However, it is not glory what they get as a reward:

> [Strickland] caught hold of the back of a chair, and, without warning, went into an amazing fit of hysterics. It is terrible to see a strong man overtaken with hysteria. Then it struck me that we had fought for Fleete's soul with the Silver Man in that room, and had disgraced ourselves as Englishmen for ever, and I laughed and gasped and gurgled as shamefully as Strickland, while Fleete thought that we had both gone mad. (Kipling 2006, pp. 13–14)

---

3    Earlier in the story, the narrator explains that he is referring to the leper as "the Silver Man" because the illness has affected the man's skin to a degree that makes it appear "like frosted silver" (Kipling 2006, p. 5).

In contrast to the heroes of imperial adventure tales, who are able to sustain their gentlemanly vigour in the face of any challenge, Strickland and the narrator do not come out of their adventure unscathed. They "disgraced [themselves] as Englishmen" in more than one way: Firstly, they accepted the authority of the "gods of the heathen" (ibid., p. 14), something which is "justly condemned" (ibid.) as the narrator concludes. Secondly, the above quotation reveals that the two agents of empire regress to an unmanly state when they are overtaken by hysteria, an illness which carried "the stigma of being a humiliatingly female affliction" (Showalter 1991, p. 106). Strickland and the narrator are unable to measure up to the ideal of imperial masculinity with its rigorous rejection of feminine qualities. In this way, British India is presented as a site of personal failure.

*3.2. Arthur Conan Doyle: "The Brown Hand" (1899)*

The second story discussed in this article is by Arthur Conan Doyle, an author who is today mainly known for the creation of Sherlock Holmes. The rest of his vast oeuvre is often overshadowed by the popular detective. Yet, Conan Doyle wrote "numerous novels and tales, plays, poems, histories, pamphlets on various issues, propaganda for spiritualism, and even a co-authored libretto" (Watts 2010, p. xii). Besides his literary career, he worked as a physician. This profession repeatedly took him to the British colonies. Although Conan Doyle defended the Second Boer War in South Africa and was an avowed Imperialist, he believed that native people should be left "unmolested and in peace" (qdt. in Stashower 1999, p. 47).

In very much the same way in which Kipling's "The Mark of the Beast" explores the idea that Englishmen might be contaminated by Eastern ideas, "The Brown Hand" deals with the anxiety that the outward movement of British imperialism could be reversed and that "familiar English spaces" might be invaded "by the body of the Other" (Macfarlane 2016, p. 82). Both stories thus dwell on imperial Gothic motifs that reinforce the Othering of the foreign culture.

In "The Brown Hand" the surgeon Sir Dominick Holden becomes haunted by the ghost of an Afghan whose hand he had amputated and kept as a fee for his services. Holden returns to England, but the phantom keeps troubling him. When he relates the encounter with the Afghan to his nephew, Dr Hardacre, who functions as the homodiegetic narrator of the story, he justifies his actions thus: "The poor man was almost a beggar, so that the idea of a fee sounded absurd, but I answered in jest that my fee should be his hand and that I proposed to add it to my pathological collection (Conan Doyle 2000, p. 78). Even though Holden emphasises that he asked for the hand "in jest", the fact that he eventually persuades the Afghan to accept his deal indicates that he considers the hand of a brown man as a valuable contribution to his pathological collection. In this way, the brown hand is both objectified and exoticised. The story thus testifies to the Othering of colonised people (and their body parts) in the British Empire.

It is further worth noting that Holden attempts to cloak his desire to possess the brown hand under a guise of generosity, for he emphasises that "the idea of a fee sounded absurd". The story thus provides an interesting allegory to British expansionism, which was often presented by its supporters as a civilising mission rather than a means to extend Britain's political and economic power. Even the Afghan's explanation that "according to his religion it was an all-important matter that the body should be reunited after death and so make a perfect dwelling for the spirit" (ibid., pp. 78–79) does not prompt Dominick Holden to change his mind. Instead, he dismisses the man's concerns as mere "superstition" (ibid., p. 79). The religious beliefs of the foreign culture are thus presented as something which must not be taken seriously.

Eventually, the Afghan accepts Holden's price. However, he warns the doctor that he shall want the hand back when he is dead—a warning which Holden ignores: "I laughed at the remark and so the matter ended" (ibid.) The fact that Holden light-heartedly rejects the Afghan's warning is further indicative of his disregard for the foreign culture.

Holden's ignorance is reminiscent of Fleete's contempt towards the Indian people in "The Mark of the Beast". It is worth emphasising that the heroes of adventure fiction often display a similar attitude.

David Ker's "A Coral Prison; Or, The Boy Hermits of the Indian Ocean", which was serialised in *The Boys Own Paper* from 4 October to 22 November 1890, serves as a good example in this context. Similar to Ballantyne's *Coral Island*, it relates the story of a bunch of boys who are shipwrecked in the Indian Ocean and subsequently marooned on an island. In the course of the narrative, the boys repeatedly violate the religious traditions of the indigenous people and penetrate sacred spaces. For instance, they ignore the native's request not to approach the tomb of a "Mohammedan saint" (Ker 1891, p. 85), who is buried on the summit of a mountain in the centre of the island. However, instead of being punished for their disrespect towards the native culture, the boys are rewarded for their behaviour. When a tsunami hits the island, they are spared from certain death only because of having climbed the sacred mountain. By contrast, the death of the native population is presented as Godly punishment. The indigenous people are referred to as "doomed" (ibid., p. 118) by the heterodiegetic narrator and the storm that precedes the tsunami is described as "the thunder of heaven" (ibid.). The religiously tinted language applied here suggests that the native population is rightly wiped from the face of the earth. "The Mark of the Beast" and "The Brown Hand" counter such stories that present the violation of the foreign religion as acceptable and even desirable by showing that the violation of religious sanctity can have fatal consequences.

In Conan Doyle's story, the Englishman realises the extent of his ignorance when the Afghan's ghost starts to haunt him. Holden understands that the spirit can only be put to rest if the Afghan is reunited with his missing hand—a seemingly impossible task, for the hand was destroyed in a fire in Holden's house in Bombay. Similar to the narrator of "The Mark of the Beast", he has to accept to supernatural power of the foreign culture that literally starts to haunt him.

In a way that conforms to Jacques Derrida's hauntological concept, the figure of the ghost suggests the return of the imperial repressed. In *Specters of Marx* (1994), Derrida notes the political and social dimension of ghosts and haunting. According to Derrida, "a specter is always a *revenant*" (Derrida 2006, p. 11). As such, it naturally disrupts the idea of a linear temporality. It draws attention to the fact that the present and the future are inevitably entangled with the past. The spectre's nature to undo "this opposition, or even this dialectic, between actual, effective presence and its others" (ibid., p. 48) is what Derrida calls the "spectrality effect" (ibid.) Building on Derrida's concept, Melissa Edmundson Makala has noted that the "idea of repetition is key to understanding the force of meaning inherent in a ghost" (Edmundson Makala 2013, p. 6). She argues that the "unsettling nature of the ghost ... exists beyond a specific moment, representing the past and influencing the future" (ibid.) The ghost is thus a constant reminder of the past. As such it carries "a political consciousness that must be recognized and dealt with" (ibid., p. 7). Just as a spectre, ideas from the past or deeds committed in the past may come back and "re-incarnate" (Derrida 2006, p. 48) themselves in the present or the future and thus remind society of its disavowed aspects. In Conan Doyle's story, the ghost of the mutilated Afghan returns to reclaim what has been taken from him by the coloniser Dominick Holden, thus reminding the doctor of his colonial deeds. The ghost's nature as a revenant underscores the idea that the postcolonial is already present in the colonial moment. As Gina Wisker stresses, "postcolonial experience is inevitably haunted by a colonial past; ... and traces of the legacy of silence, pain, humiliation, and dispossession reappear in spectral figures" (Wisker 2016, p. 511).

Four years have passed when Dominick Holden confides in his nephew. By this time, the nightly appearance of the ghost has left its mark on the surgeon:

> His figure was the framework of a giant, but he had fallen away until his coat dangled straight down in a shocking fashion from a pair of broad and bony shoulders. ... [T]he appearance and bearing of the man were masterful, and one expected a certain corresponding arrogance in his eyes, but instead of that I read the look which tells of a spirit cowed and crushed, the furtive, expectant look of a dog whose master has taken the whip from the rack". (Conan Doyle 2000, p. 73)

In contrast to imperial adventure tales, where the young hero engages in a dangerous journey in the colonies and returns to the United Kingdom a better and stronger man, Dominick Holden's return

is marked by decay. The way the coat dangles down from Holden's shoulders is described by the narrator as "shocking", thus indicating his bodily regress. Holden's physical weakness is accompanied by a "look which tells of a spirit cowed and crushed". The surgeon's regress is thus presented as both a physical and a mental process. Akin to the Afghan's ghost, Dominick Holden is reduced to the ghost of his former self. Considering prevailing anxieties about the future of the colonial project, it is possible to read Holden's regress as a metaphor for the crumbling British Empire that is being eroded by foreign powers and starts to wither like the old man in Conan Doyle's narrative. Simon Hay similarly argues that the surgeon's depiction "is indicative of a whole set of imperial anxieties, in which Holden's body stands in allegorically for the British Empire" (Hay 2011, p. 136).

The narrative further "articulates profound anxieties about Western scientific practices" (Macfarlane 2016, p. 82). The encounter with the Afghan is related by Holden himself. Hence the reader cannot be certain whether the amputation was necessary in the first place. Just as the Afghan man himself, we can only take Dominick Holden's word for it: "He was suffering from a soft sarcomatous swelling of one of the metacarpal joints, and I made him realise that it was only by losing his hand that he could hope to save his life" (Conan Doyle 2000, p. 78), he relates. There is nobody who could confirm the doctor's diagnosis. The story thus opens up a space for speculations about the rightfulness of the amputation. By implying that the operation might not have been necessary, the story again points to the dehumanisation and objectification of the foreign Other, whose bodily integrity is subordinated to Holden's scientific interests.

The Afghan's "position as undifferentiated object of imperial intervention continues" (Macfarlane 2016, p. 83) as the story progresses. In an attempt to satisfy the restless spirit, Holden's nephew organises a brown man's hand and puts it in one of the glass jars that Holden uses for his pathological collection. However, his first attempt fails because he did not even think of getting the correct hand. While the Afghan's ghost is missing his right hand, Hardacre took the left hand of a dead Indian man from a British hospital. Eventually, he succeeds in his mission. Once the ghost finds the right hand of another brown man in the jar, he is satisfied and disappears. By suggesting that any brown hand will do, the body of the colonised Other is again objectified. In addition, the story implies the superior intelligence of the white doctor, who can trick the Afghan, who, in turn, is depicted as stupid and easily satisfied. To a certain extent, the story thus rehabilitates the image of the superior, middle-class Englishman.

The revitalisation of white, middle-class masculinity is also implied by Holden's recovery at the end of the story. Like Fleete in "The Mark of the Beast", Dominick Holden regains his strength when the haunting is lifted. Both stories thus suggest that revitalisation is possible. Nevertheless, the men's temporary regress blurs the binary divisions between West and East, civilised and savage, ruler and subject and thus exposes such dualisms as social constructs.

## 4. Conclusions

To my knowledge, this article constitutes the first comparative reading of Kipling's "The Mark of the Beast" and Conan Doyle's "The Brown Hand". Both authors have written several Gothic stories and tales of the supernatural. So far, however, this aspect of their writing has often been neglected in academia. This article has shown that Kipling's and Conan Doyle's Gothic writing offers nuanced commentaries on political, cultural, and social issues and is therefore deserving of further criticism.

"The Mark of the Beast" and "The Brown Hand" are not only Western texts that replicate the stereotypical depiction of the East as savage and monstrous. Both stories expose the consequences of British imperialism—for the colonised people and the British intruders. They do so by revealing the cruelty of Britain's imperial practices. This cruelty is suggested by the torture scene in Kipling's "Mark of the Beast". It also underlies the doctor's insistence on keeping the hand of a poor Afghan man in Conan Doyle's "The Brown Hand". Furthermore, these stories imply that the agents of Empire do not remain unpunished. The imperial repressed return to haunt their oppressors. In both narratives, the supernatural punishment is self-inflicted. The white colonisers Fleete and Dominick Holden are

shown as responsible for their misery. Their feeling of superiority and their ignorance towards the traditions of the foreign culture are identified as highly problematic. "The Mark of the Beast" and "The Brown Hand" thus offer what Stephen Arata calls "responses to cultural guilt" (Arata 1990, p. 623). It is therefore justified to read these two stories through a postcolonial lens. I do not mean to suggest that these stories fully grasp the extent of the violation experienced by the colonised people through Britain's interference with their culture. Being written from a quintessentially British perspective, they could never do so. Nevertheless, they erode the image of the clever, courageous, and cool-headed hero, as well as the model of imperial masculinity that was so carefully constructed in adventure fiction and dismantle the duality between civilised Western self and barbaric Eastern Other as an imperialist myth.

**Funding:** This research received no external funding.

**Conflicts of Interest:** The authors declare no conflict of interest.

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
