# Peer review of "Haunted Oppressors: The Deconstruction of Manliness in the Imperial Gothic Stories of Rudyard Kipling and Arthur Conan Doyle"

_humanities, doi:10.3390/h9040122_

Round 1

Reviewer 1 Report

This is a well-paced, well-structured article. It makes clear thesis statements, reinforced through ample use of evidence that is well referenced. It makes good use of subtitles to break off distinct areas of the discussion, though I would suggest considering an additional subtitle around line 284 to denote where the analysis moves on to the second text. 

The argument is clearly stated in the introduction, and backed up by a perceptive choice of texts. Though some of the analysis of Kipling's 'Mark of the Beast' does not feel like its covering much new ground it serves as a very clear example of the article's wider point. The discussion of Kipling in a Gothic context is something that is an area deserving of further study. The exploration of the Conan Doyle short story is much needed, and one of the strengths of the piece.

Around line 321, the remark about adventure fiction's heroes displaying similar attitudes, may benefit from an example, or additional foregrounding during the adventure fiction section for later ones to call back to. The adventure fiction section does a great job in outlining the outlook of the subgenre as a whole, but does not go into the specifics that the article does with its own examples - this limits the comparison at times. 

Just a suggestion for line 381: I'd considering moderating the tone a little - does the story suggest white intelligence, or is it just blind/ignorant to this problematic implications of this deception?

I also wonder is line 397's 'On the other hand' is a deliberate pun? If this is unintentional it may be worth revising.

As the conclusion is quite brief, I believe there is room for some expansion here. As it stands, this conclusion feels quite a brief afterword on a complex, potentially sensitive, topic - it feels more like a summary of your analysis of the two texts rather than a fitting response to the claims you make in the introductory paragraphs. You will have better ideas about how to unpack this conclusion a little more, but I'd give consideration to going into further detail in the adventure fiction section that you could then respond to in the conclusion - this would allow you to draw more general conclusions about the two forms. On a more specific note I'd suggest reframing the remarks that follow 'I am not suggesting' to allow you to more confidently assert that these texts show Gothic Imperialism's capacity to 'erode the image of the clever, courageous' etc. 

Author Response

This is a well-paced, well-structured article. It makes clear thesis statements, reinforced through ample use of evidence that is well referenced. It makes good use of subtitles to break off distinct areas of the discussion, though I would suggest considering an additional subtitle around line 284 to denote where the analysis moves on to the second text. 

Response: There is already a subtitle at the beginning of the second text. The reviewer probably overlooked the subtitle because it was situated at the very end of page 6 instead of the beginning of page 7. In the reworked version the subtitle is situated on page 7, line 283.

The argument is clearly stated in the introduction, and backed up by a perceptive choice of texts. Though some of the analysis of Kipling's 'Mark of the Beast' does not feel like its covering much new ground it serves as a very clear example of the article's wider point. The discussion of Kipling in a Gothic context is something that is an area deserving of further study. The exploration of the Conan Doyle short story is much needed, and one of the strengths of the piece.

Response: Thank you for your constructive feedback. I have included a comment which states that Kipling’s and Conan Doyle’s Gothic texts are deserving of further study (ll. 412-16).

Around line 321, the remark about adventure fiction's heroes displaying similar attitudes, may benefit from an example, or additional foregrounding during the adventure fiction section for later ones to call back to. The adventure fiction section does a great job in outlining the outlook of the subgenre as a whole, but does not go into the specifics that the article does with its own examples - this limits the comparison at times. 

Response: Thank you for this helpful suggestion. I have included an example, see ll. 322-344

Just a suggestion for line 381: I'd considering moderating the tone a little - does the story suggest white intelligence, or is it just blind/ignorant to this problematic implications of this deception?

Response: I understand your concern. However, I would still argue that the text presents the Afghan as clearly inferior to and less intelligent than the British doctor.

I also wonder is line 397's 'On the other hand' is a deliberate pun? If this is unintentional it may be worth revising.

Response: No, it was not deliberate. Thank you for pointing out that it might come across as macabre to use the phrase in this context. I have changed “On the other hand” to “Furthermore”, l. 424.

As the conclusion is quite brief, I believe there is room for some expansion here. As it stands, this conclusion feels quite a brief afterword on a complex, potentially sensitive, topic - it feels more like a summary of your analysis of the two texts rather than a fitting response to the claims you make in the introductory paragraphs. You will have better ideas about how to unpack this conclusion a little more, but I'd give consideration to going into further detail in the adventure fiction section that you could then respond to in the conclusion - this would allow you to draw more general conclusions about the two forms. On a more specific note I'd suggest reframing the remarks that follow 'I am not suggesting' to allow you to more confidently assert that these texts show Gothic Imperialism's capacity to 'erode the image of the clever, courageous' etc. 

Response: Thank you for your suggestion. I have expanded on the conclusion. However, with regards to the word limit, I could only add a few extra lines. See ll. 412-417 and ll. 429-430.

Reviewer 2 Report

This article offers a clear and persuasive thesis statement and at times demonstrates impressive nuance in articulating the argument under discussion - the justification of the situating of the two short stories at the borderline between Imperial Gothic and Postcolonial Gothic is one such thoughtful move.The work is well researched (indeed at times the author would do well to allow their own critical voice to come more to the fore rather than relying upon existing criticism), and clearly structured. The 'Gothic' status of the stories under discussion is perhaps a little taken for granted - attention to the omission of the torture scene in 'Mark of the Beast' and its relationship to the Gothic motif of the incomplete document or the story which refuses to tell (is, indeed, 'unspeakable') would help to mitigate this.

There are three key areas where the article could be strengthened. The first is in staking its claim to critical originality. At present nowhere in the paper does the author articulate the new ground the work is breaking or where it deviates from established critical understandings of the short stories. From looking at the literature it appears that these two short stories are very infrequently read together and even the Macfarlane article which is cited here, and offers the closest area of critical concern to that under discussion here, does not offer the attentive and comparative reading of the stories which is undertaken by the author so there are claims for originality which could be made but this needs to be explicitly articulated.

The second area for attention is in terms of the written expression. There are numerous moments when the prose requires further sophistication and precision. For example, speaking about 'the Indian culture and religion' (ll. 33-4) is not sufficient given the profoundly culturally and religiously diverse character of India as a country. More confident control of key terms would also strengthen the piece - 'Gothic literature' and 'supernatural literature' appear to be used synonymously which is unhelpful. An articulation both of the shared territory those terms cover and of the specificity of the terms would have enhanced the precision of the readings. Finally more careful proof-reading will flag up sentence fragments and remnants from previous drafts or guidance documents (lines 151-3 appear to have been left in from an earlier guidance document for example).

The third element of the article which would benefit from attention is the deployment of the Derrida. While the author makes no inaccurate statements about Derrida's theory of spectrality and the spectre, the use of his work here feels unhelpfully general. The author may wish to consider recognising the significance of debt and obligation to Derrida's model of spectrality, and of the profoundly ethical implications of his model, in their use of these concepts to read the primary texts.

Author Response

There are three key areas where the article could be strengthened. The first is in staking its claim to critical originality. At present nowhere in the paper does the author articulate the new ground the work is breaking or where it deviates from established critical understandings of the short stories. From looking at the literature it appears that these two short stories are very infrequently read together and even the Macfarlane article which is cited here, and offers the closest area of critical concern to that under discussion here, does not offer the attentive and comparative reading of the stories which is undertaken by the author so there are claims for originality which could be made but this needs to be explicitly articulated.

Response: Thank you for your suggestions. I have included a paragraph which states that, to my knowledge, the article constitutes the first comparative reading of the two stories. I have also emphasised that Kipling’s and Doyle’s Gothic writing is rarely discussed in academia and that my article shows that it is deserving of further research. See ll. 412-17.

The second area for attention is in terms of the written expression. There are numerous moments when the prose requires further sophistication and precision. For example, speaking about 'the Indian culture and religion' (ll. 33-4) is not sufficient given the profoundly culturally and religiously diverse character of India as a country. More confident control of key terms would also strengthen the piece - 'Gothic literature' and 'supernatural literature' appear to be used synonymously which is unhelpful. An articulation both of the shared territory those terms cover and of the specificity of the terms would have enhanced the precision of the readings. Finally more careful proof-reading will flag up sentence fragments and remnants from previous drafts or guidance documents (lines 151-3 appear to have been left in from an earlier guidance document for example).

Response: Thank you for your observations. I added a footnote in which I comment on the difference between the terms Gothic literature and supernatural literature (footnote 2, p. 2). I also agree that “Indian culture and religion” is not specific enough and changed it to “culture and religion of Hindu people in North India” (see ll. 33-35). I also changed “branding the Hindu deity” to “branding the statue of the Hindu deity” (l. 191). Furthermore, I added a footnote which states that Kipling spells the city of Dharamsala “Dharmsala” in his story (see footnote 1, p. 1) and a footnote that explains why the narrator of “The Mark of the Beast” refers to the leper as “the Silver Man” (footnote 3, p. 6)

I have proof-read the article, again, and hope that there are no more typos, remnants from previous drafts etc.
Mistakes that I have corrected:
- I deleted “of West Africa” in l. 133 because it is repetitive
- writings to writing (l. 164)
- p. 3 instead of p 33 (l. 173)
- cultures to culture (l. 218)
- English to British (l. 220)
- I added a comma in l. 227
- “the English surgeon” to him because it was repetitive (l.299)
- spectre to specter in derrida’s quote (the book was translated into AmE, see l. 347)
- I added a comma in l. 348

The third element of the article which would benefit from attention is the deployment of the Derrida. While the author makes no inaccurate statements about Derrida's theory of spectrality and the spectre, the use of his work here feels unhelpfully general. The author may wish to consider recognising the significance of debt and obligation to Derrida's model of spectrality, and of the profoundly ethical implications of his model, in their use of these concepts to read the primary texts.

Response: Thank you very much for pointing out the significance of Derrida’s concept. I have extended the section on Derrida, see ll. 345-46 and ll. 356-58. However, a more nuanced reflection of his theory would, unfortunately, exceed the word limit I have been given for this article – also because a more intensive discussion of Derrida would entail another paragraph that deals with the criticism Derrida has received for Specters of Marx (by Gayatri Chakravorty Spivak, for instance). I hope that you find my discussion of Derrida’s concept improved.